# Recent Advancements and Future Perspectives of Microalgae-Derived Pharmaceuticals

**DOI:** 10.3390/md19120703

**Published:** 2021-12-12

**Authors:** Donghua Xia, Wen Qiu, Xianxian Wang, Junying Liu

**Affiliations:** 1State Key Laboratory of Food Science and Technology, The Engineering Research Center for Biomass Conversion, Nanchang University, Nanchang 330047, China; xdonghua@ncu.edu.cn; 2Eco-Environmental Protection Research Institute, Shanghai Academy of Agricultural Sciences, Shanghai 201403, China; wqiu2015@zju.edu.cn; 3Institute of Toxicology and Genetics, Karlsruhe Institute of Technology, 76344 Eggenstein-Leopoldshafen, Germany; xianxian.wang@kit.edu; 4Pharmaceutical Manufacturing Technology Centre (PMTC), Bernal Institute, University of Limerick, V94T9PX Limerick, Ireland

**Keywords:** pharmaceuticals, polyunsaturated fatty acids, polysaccharides, astaxanthin, beta-glucan, immune activation

## Abstract

Microalgal cells serve as solar-powered factories that produce pharmaceuticals, recombinant proteins (vaccines and drugs), and valuable natural byproducts that possess medicinal properties. The main advantages of microalgae as cell factories can be summarized as follows: they are fueled by photosynthesis, are carbon dioxide-neutral, have rapid growth rates, are robust, have low-cost cultivation, are easily scalable, pose no risk of human pathogenic contamination, and their valuable natural byproducts can be further processed. Despite their potential, there are many technical hurdles that need to be overcome before the commercial production of microalgal pharmaceuticals, and extensive studies regarding their impact on human health must still be conducted and the results evaluated. Clearly, much work remains to be done before microalgae can be used in the large-scale commercial production of pharmaceuticals. This review focuses on recent advancements in microalgal biotechnology and its future perspectives.

## 1. Introduction

Interest in microalgae bioactive compound production is being driven by a growing number of studies demonstrating their beneficial effects on human health [1]. These algae-based bioactive compounds are essential for the development of drugs for human diseases, as many viable drugs are microalgae-originated products and often exhibit antitumor and anti-inflammatory activities [2]. As fast-growing and solar-powered biofactories, microalgae are ideal platforms that possess the potential to meet the enormous demand of biopharmaceutical and human nutraceutical production. Moreover, microalgal cell factories provide a wide range of pharmaceutical products, recombined proteins, vaccines, and drugs that otherwise are unavailable or are too expensive to produce from human cell lines or plant sources [3] (Figure 1). Compared to higher plants, the advantages of microalgae include their (1) higher production yields of bioactive compounds through solar energy and carbon dioxide utilization; (2) shorter growth periods with a doubling time <24 h, especially for large-scale production [4]; (3) no competition for or need to occupy land areas; (4) relatively simple supplementation of climate conditions; and (5) controllable bioactive compound production by regulating growth conditions. Additionally, most algae have been granted the Generally Recognized as Safe (GRAS) status, which makes the use of microalgae as cell factories very appealing for pharmaceutical and food purposes. Although the commercialization of various high-value pharmaceutical products from microalgae is still in its infancy, it will likely become a gateway to a multibillion-dollar industry in the near future [3].

Bioactive compounds have been the subject of previous reviews that have focused on algae-based dietary supplements [1], recombinant proteins [5], and algae-derived anticancer drugs [6]. This review will focus on recent advancements in naturally derived valuable compounds and recombinant proteins explicitly used for pharmaceutical applications. The first section details the current status of these naturally derived bioactive compounds, including polyunsaturated-eicosapentaenoic acid (C20 𝜔-3:5, EPA), docosahexaenoic acid (𝜔-3 C22:6, DHA), polysaccharides, and astaxanthin, which are discussed in terms of their structure/property relationships. The second section describes recombinant proteins. The third section outlines the main technological hurdles for microalgal cell factories. The fourth section summarizes a description of the latest developments related to genetic and metabolic engineering. The fifth section is about the latest and prospective applications of algal cell factories discussed in detail.

## 2. Bioactive Compounds

It has been well established that microalgae produce a wide range of primary and secondary metabolites that possess antifungal, antiviral, immune activation, or antibiotic properties, which can be potentially exploited by pharmaceutical and nutraceutical sectors on a large commercial scale [3]. These bioactive molecules may be obtained directly from primary metabolites, such as polyunsaturated fatty acids (PUFAs) and polysaccharides or synthesized from secondary metabolites (Figure 1).

### 2.1. PUFAs

PUFAs are the most studied microalgal lipid compound. These renewable bioactive lipids have been used in the prevention and treatment of cardiovascular diseases (CVDs) [7]. A previous study showed that dietary 𝜔-3 PUFAs have a protective effect against atherosclerotic heart disease [8]. Derivatives of PUFAs, namely DHA, EPA, α-linolenic acid (ALA), and docosapentaenoic acid, have been applied to treat type 2 diabetes, inflammatory bowel disorders, skin disorders, and asthma [8]. DHA and EPA have been demonstrated to reduce medical complications associated with strokes and arthritis, and reduce hypertension while acting as anti-inflammatory agents [2]. Additionally, DHA is important for the development and function of the nervous system. Microalgal phytosterols have interesting bioactive properties, such as reducing cholesterol (low-density lipoprotein) in humans by inhibiting its absorption from the intestine. The regular consumption of DHA and EPA supplements can reduce the risk of CVDs [9]. Adequate intake of DHA and EPA during pregnancy is also crucial for the healthy development of the fetal brain [7].

Sufficient intake of arachidonic acid (ARA) and DHA is essential for the functional development of the infant and to ensure average intrauterine growth [9]. Furthermore, ARA and EPA are platelet aggregators and vasodilators, and have antiaggregative effects on chemostatic activity and neutrophils. Unsurprisingly, the demand for microalgal-derived PUFAs, such as ARA and DHA, which have been added as fortifications to infant formula, has created an industry worth USD 10 billion per year [9]. Research has shown that an extract from *Nannochloropsis oceania* protects neuronal cells and inhibits oxidative stress caused by β-amyloid (Aβ). Several other studies with epidemiological evidence have shown that PUFAs can reduce the risk of Alzheimer’s disease. DHA can also ameliorate synatosomal membrane fluidity, as well as prevent Aβ production and aggregation, in APP/PS1 transgenic rat brains by modulating amyloid preselinin protein (APP) processing or by modulating the fibrillar oligomer [10]. Therefore, DHA/EPA could help vaccines target certain inflammatory disorders. Moreover, DHA/EPA could also agitate the immune system based on different cell types or target species by promoting β-cell activation, thereby providing protection against infectious diseases.

In animal cells, PUFAs, especially DHA and EPA, are important brain and eye development components and cardiovascular vessels [11]. However, DHA is not synthesized and is acquired through the diet; fish and marine microalgae are the primary sources of 𝜔-3 PUFA. Fish tend to obtain EPA through the bioaccumulation of EPA in the food chain, while microalgae produce 𝜔-3 PUFA [11]. However, since global fish stocks are declining due to overfishing, their derived oils have been increasingly contaminated with heavy metals and toxins. Fish oil cannot meet the growing demand for purified PUFAs, making DHA/EPA production from microalgal biotechnology seem an attractive alternative (Table 1).

Microalgae offer the following advantages compared to fish oil: (1) the PUFA profile is more straightforward; (2) cultivation conditions can be controlled; (3) different algal species produce different PUFAs; (4) fatty acid production from microalgae has the potential to be less expensive than fish; (5) microalgal fatty acids have no odor, metal contamination, or cholesterol; (6) EPA esters obtained from microalgae are more stable than from fish oil products [11]; and (7) microalgal lipid levels usually range from 20–50%, but can accumulate up to 80%, of the DW under stressful conditions, such as nitrogen starvation or elevated temperatures [11,17]. For example, *Phaeodactylum tricornutum* produces mainly EPA; *Nitzschia conspicua* produces arachidonic acid (𝜔-6 C20:4, or ARA, an essential fatty acid); and *Schizochytrium* sp. accumulates DHA, EPA, and palmitic acid, which may account for 56% of the total dry weight (DW) [18] (Figure 2). For these reasons, algae are garnering increasing attention in the healthcare and biomedical fields. Interestingly, besides provide extraction of PUFAs, the addition of red algal extracts from Gracilaria chilensis, *Gelidium chilense*, *Iridaea larga*, *Gigartina chamissoi*, *Gigartina skottsbergii*, and *Gigartina radula* prevents the degradation of almost a half of DHA and 25% of EPA upon cooking heat treatment, and it also inhibits the development of diverse pathogenic bacteria (*Bacillus cereus*, *Escherichia coli*, *Staphylococcus aureus*, *Pseudomonas aeruginosa*, *Proteus mirabilis*, and *Salmonella enteritidis*) [19].

To produce high PUFA yields, different extraction techniques as well as various solvents and solvent systems have been evaluated. Pressurized liquid extraction (PLE, also known as accelerated solvent extraction) and microwave-assisted solvent extraction (MAE) techniques were compared for PUFA, particularly EPA extraction in *Phaeodactylum tricornutum* [20]. The PLE method provides higher extraction yields although the MAE technique results in extracts with higher antioxidant activity, which may be caused by the different optimized extraction temperatures (50 °C and 30 °C for PLE and MAE, respectively) [20]. Moreover, EPA extraction was determined using the tight, thick, and complex multilayered microalgal structure of *Nannochloropsis oceanica*. As a result, the EPA contents extracted from spray-dried biomass using different solvent systems were ranked as follows: dichloromethane/methanol (DM) ≈ chloroform/methanol (CM) > ethanol extraction assisted with ultrasound probe (USP) > ethanol ≈ dichloromethane/ethanol (DE) > ethanol extraction assisted with ultrasound bath [21]. DM, CM, and ethanol with USP solvents extracted similar yields of EPA, and with assistance of USP, ethanol extraction leads to about a 35% increase in EPA content compared to ethanol alone [21].

### 2.2. Polysaccharides

Polysaccharides are biopolymers in which ≥10 monosaccharides are linked in linear or branched chains via glycoside bonds. There are a growing number of concerns regarding polysaccharides as high-value components in pharmaceutical, food, cosmetic, fabric, and emulsifier applications. Sulfated polysaccharides from microalgae contain sulfate esters and possess attractive pharmaceutical applications, including antioxidant, anticoagulant, anti-inflammatory, antiviral, antibacterial, antitumor, immunomodulatory, and radioprotective properties [6]. These compounds can stimulate macrophage modulation, and their bioactive action is based on their sugar composition, structure, and degree of sulphation. The sulfated polysaccharides from microalgae exhibit high antiviral activity and are mainly composed of xylose, glucose, and galactose. The antiviral mechanism is based on the inhibitory effects elicited by positive charges on the cell surface, thereby preventing viruses from entering the host cell. Algal sulfated polysaccharides inhibit various viruses from HIV to swine flu [22], similar to the inhibitory effects of carrageenan on the human papillomavirus [23].

Sulfated polysaccharides from the cyanobacteria *Spirulina* have been used as antiviral drugs both in vivo and in vitro, while polysaccharides from eukaryotic microalgae, such as *Chlorella vulgaris*, *Scenedesmus quadricauda*, and *Dunaliella* sp., have high inhibitory viral activity [24]. The p-KG103 derived from the marine microalga *Gyrodinium impudium* can prevent tumor cell growth by stimulating cytokine production [25]. The sulfated polysaccharide from *Tribonema* sp. is a heteropolysaccharide mainly composed of galactose, and exhibits immunomodulatory activity through the stimulation of macrophage cells by upregulating IL-6, IL-10, and tumor necrosis factor α [26]. The mechanism of this polysaccharide’s anticancer action is the induction of cell apoptosis, rather than affecting the cell cycle or mitosis of HepG2 cells. Soluble sulfated polysaccharide fractions can be released into the surrounding medium by the red microalgal marine species *Porphyridium* sp. The freshwater alga *P. aerugineum* lacks the rigid microfibrillar components that are typical in most algal species. The soluble sulfated polysaccharide fractions are classified as anionic heteropolymers based on the presence of glucuronic acid residues and sulfates containing 10 different monosaccharides—mainly xylose, glucose, and galactose. With a high molecular weight in the order of 4810 kDa, this polysaccharide has displayed anticancer activity in a dose-dependent manner (up to 60.37% at 250 µg/mL) as evaluated by MTT assays, which makes it a potentially attractive drug for future anticancer treatment [24]. Polysaccharides with low molecular weight fragments derived from *P. cruentum* and *Spirulina* sp. have antioxidant activity; additionally, their free radical-scavenging capacity was inferred to be associated with the total phenolic content [27].

Polysaccharides are important for regulating immune responses [26,28]. For example, *C. vulgaris* polysaccharides have displayed high immunomodulatory activity [29]. Therefore, extracellular polymeric substances containing polysaccharides from *C. vulgaris* are a promising agent for chronic airway inflammation prevention, the primary pathogenic mechanism of many respiratory diseases, including bronchial asthma (Table 2). Furthermore, this natural compound can improve the overall survival rate of the nematode *Caenorhabditis elegans* under increased oxidative stress induced by paraquat. These data suggest that nostoglycan can reduce reactive oxygen species levels, inhibit protein carbonyl formation, and improve the activities of superoxide dismutase and catalase in paraquat-exposed nematodes. Accordingly, it was inferred that polysaccharides can potentially suppress tumor proliferation and induce lung apoptosis [30]. The proposed mechanisms underlying its antitumor effect include: (1) the induction of apoptosis of tumor cells or inhibition of the expression of cellular oncogenes to kill tumors directly; (2) the improvement of host immune function; and (3) synergistic effects with some traditional chemotherapy drugs. Polysaccharides from other microalgal species also display interesting bioactivity. For instance, a polysaccharide fraction (HPP-c3-s1) obtained from *Haematococcus pluvialis* is a pyranose containing amino and O-acetyl groups, and shows substantial immunostimulatory effects on splenocytes and B lymphocytes [31]. Antiaging assays also showed that HPP-c3-s1 can extend the mean survival of nematodes. A water-soluble polysaccharide extracted from *N. oculata* is rich in (β1→3, β1→4)-glucans, (α1→3)-, (α1→4)-mannans, and anionic sulfated heterorhamnans, which can stimulate murine B-lymphocytes, as evaluated by an immunostimulatory assay. However, the mechanism underpinning their biological activity requires further investigation [31].

Biological functions of sulfated polysaccharides depend on the presence and spatial positioning of sulfo groups (sulfate content), molecular weight, fucose content, and polyphenols [32]. For polysaccharides, sulfated modification enables/potentiates their antioxidant activity, and the disappearance of antiviral bioactivities is highly associated with the removal of sulfated groups [51], the anticoagulant activity of full or partially desulfated polysaccharides is lost compared to the native polysaccharide [52]. For instance, 6-O-desulfonated, 1→3 linked polysaccharides, e.g., curdlan and galactan, lost 50% of the anticoagulant activity [53]. Moreover, sulfated polysaccharides with higher molecular weight show better antibioactivity. One example is the fully sulfated galactan with the smallest molecular weight which showed the lowest antithrombin activity compared to xylan, amylose, cellulose, and curdlan [53]. With a similar 27% sulfate content, 735 kDa of fucoidan from the seaweed *Fucus vesiculosus* results in IC_50_ at 0.35 μg/mL [32], and 34.4 kDa of fucoidan from seaweed *Ascophyllum nodosum* causes 30.4% scavenging of DPPH at 10 mg/mL [33] (a lower IC_50_ value corresponds to a stronger antioxidant activity of the sample). However, there is also controversy about low molecular weight polysaccharide LBP from *Lilii Bulbus* which shows better immunomodulatory effects than those with higher molecular weight [54]. λ-carrageenan from *Chondrus ocellatus* with molecular weights of 9.3 and 15 kDa showed better antitumor and immunomodulation activities [55]. Fucoidans are valuable to maintain oral hygiene and prevent dental caries. Fucoidans show strong antimicrobial activity against the Gram-negative bacterium *Porphyromonas gingivalis* as well as Gram-positive bacteria *Streptococcus mutans* and *Candida albicans*, either through specific binding to pathogens that can directly neutralize endotoxin or using an unknown mechanism [36]. Moreover, crude fucoidans show a stronger inhibitory effect on adhesion to teeth than pure fucoidans [36].

The extraction method affects the yields of sulfate polysaccharide compounds and antioxidant properties [56]. The author and colleagues evaluated different technologies for extraction of sulfate polysaccharide, including microwave-assisted extraction (MAE), hot-water extraction, and ultrasound-assisted extraction. With the same ratio of water to raw material, MAE extracts had the highest polysaccharide yield (9.6%), while the hot-water extracts had the lowest polysaccharide yield (8.5%) for *Gracilaria lemaneiformis* [37]. MAE may offer a rapid delivery of energy to the solvent and thus microwave radiation can be focused directly onto the sample, therefore, the heating is more efficient with shorter processing time (20 min vs. 60 min for hot-water extraction) [37].

Extraction temperature plays an important role in sulfate polysaccharide extraction, as the molecular weight, sulfate content, and antioxidant activities of extracted fucoid varied with extraction temperature using the MAE method [33], probably because of the difference in the produced monosaccharide composition: fucose at 90 °C and fucoidan at 150 °C for *Ascophyllum nodosum* [33]. The highest fucoidan yield (16.1%) was obtained using MAE at 120 °C for 15 min [33]. Although the yield of fucoidan was 21.0% using conventional hotplate heating for 9 h, the balance between yield and extraction time needs to be considered [33]. In addition, the pressure used for MAE extraction strongly influences the fucoidan composition: galactose contents increased with the increase in pressure from 30 psi to 120 psi, and fucose was only present at 30 psi while xylose was only present at 120 psi for seaweed [57]. Techniques are developing towards increasing the efficiency of extraction. Recently, researchers reported 10-fold higher yields of fucose-sulfated polysaccharide (FBP) extracts from *A. nodosum* using MAE, although antioxidant activities of extracts were slightly improved using the ultrasound extraction technique. Furthermore, the simultaneous application of microwaves and ultrasounds causes significant yield increases in the extraction of FBPs compared to single utilization and further confirmation is needed to test the antioxidant properties of these extracts in other biological models [56].

### 2.3. Astaxanthin

Astaxanthin is an orange-red, fat-soluble pigment with the molecular formula C_40_H_52_O_4_ [58]. Astaxanthin belongs to the carotenoid xanthophyll class, which includes β-cryptoxanthin, canthaxanthin, lutein, and zeaxanthin. Astaxanthin is amphipathic, allowing it to be carried by lipid molecules directly to tissues and organs (Figure 3). It can also be transported into the liver via the lymphatic system but cannot be converted into vitamin A in the liver. Astaxanthin consists of three isomers: 3S,3′S, 3S,3′R, and 3R,3′R [59]. The 3S,3′S isomer has the most potent antioxidant properties, facilitating cell membrane penetration while maintaining cell membrane integrity. The unique chemical structure of astaxanthin allows it to cross the blood–brain barrier, which could likely play an essential role in treating central nervous system diseases [59]. Additionally, astaxanthin exhibits various biological characteristics, such as anti-inflammation, antioxidation, and antiapoptosis activities [10]. Natural 3S,3′S derived from *H. pluvialis* exhibits 14–65 times higher antioxidant activity than vitamins C and E, as well as 20 times more efficient antioxidant activity than synthetic astaxanthin [60].

Astaxanthin can be produced from red yeast (*Phaffia rhodozyma*), green algae (*H. pluvialis* and *C. zofingiensis*), and marine organisms such as salmon and lobsters [61]. However, only those derived from *H. pluvialis* or *Paracoccus carotinifaciens* have been authorized for direct human consumption (in the form of dietary supplements) by the US Food and Drug Administration (FDA) at dosages up to 12 mg/day for no more than 30 days and up to 24 mg/day at a dose of 6 mg/day [60]. Moreover, astaxanthin derived from *H. pluvialis* has been authorized for direct human consumption by the European Food Safety Authority, both in dried form or with an ethanol or supercritical CO_2_ extraction carrier [60]. Astaxanthin derived from other sources has been approved globally for use in salmon and trout feeding under different conditions [60]. Still, most astaxanthin supplements for farmed salmon use 3R,3′S or 3R,3′R isomers from yeast [60].

Antioxidants can slow down oxidative stress processes that damage membrane lipids, proteins, and DNA, thereby alleviating many health disorders, such as certain cancers, diabetes, and some neurodegenerative and inflammatory diseases related to oxidative stress [27]. For the applications of astaxanthin’s antioxidant potential in humans, please refer to reference [62]. Although the mechanisms mediating the antioxidative action of astaxanthin have yet to be elucidated, several molecular targets of astaxanthin have been proposed, which may explain the biological effects of its natural pharmaceutical properties. The intense antioxidative activity of astaxanthin makes it an attractive multi-target pharmacological agent for protection against inflammation, oxidative stress, and apoptosis, which play critical roles in the pathogenesis of many chronic neurological diseases and CVDs in humans, such as Alzheimer’s disease, inflammatory injuries, age-related dementia, myocardial infarction, localized cerebral ischemia, and peripheral vascular disease (Table 3).

(1)Astaxanthin may alleviate neurodegenerative injuries in Alzheimer’s disease. One of the most common forms of dementia, Alzheimer’s disease is widely spread and cannot be ignored [63]. More than 46 million humans currently live with dementia, and over 130 million estimated cases are expected worldwide by 2050 [10]. While the etiology of Alzheimer’s disease is not fully understood [10], astaxanthin exhibits neuroprotective effects in retinal ganglion cells and can regulate the AKT/GSK-3b signaling pathway [76]. Collectively, these properties, along with the antioxidative activity of astaxanthin, may protect against Alzheimer’s disease.(2)Astaxanthin performs multiple biological activities. For example, it can alleviate neurotoxicity in young and aged rat models of Parkinson’s disease (PD) after exposure to the neurotoxin MPTP through which neurons in the substantial nigra of aged mice groups are preserved [66]. Moreover, astaxanthin has more potent effects in young animals, as the loss of tyrosine hydroxylase through the nigrostriatal circuit induced by MPTP in aged mice cannot be reduced.(3)Researchers have suggested that astaxanthin may enhance the expression of a brain-derived neurotrophic factor in rats with a subarachnoid hemorrhage (SAH), a serious clinical disease that makes its victims comatose. In contrast, neuronal differentiation factors were reduced after SAH [81]. Although the mechanism underlying clinical symptoms is still not fully understood, the mechanism underlying the inhibition of astaxanthin in mitochondria-associated neuron apoptosis has been proposed to be (1) increased mitochondrial membrane potential; (2) decreased Bax/Bcl-2 ratio; (3) inhibition of the release of cytochrome C to the cytoplasm; and (4) suppression of caspase-3 enzyme activity [79].(4)Astaxanthin exhibits protective activity in the central nervous system by inhibiting neuronal damage induced by H_2_O_2_. Additionally, astaxanthin may act as a neuroprotective agent against spinal cord injury-induced neuronal damage by attenuating oxidative damage and inhibiting apoptosis [77]. Furthermore, it has been confirmed that astaxanthin can reduce the effects of ischemic brain injury in adult rats by inhibiting glutamate overflow. However, the direct and indirect effects of astaxanthin on the expression of aquaporins and Na^+^-K^+^-2Cl^−^ co-transporters during traumatic brain injury remain to be determined [73]. The proposed underlying mechanism is as follows: (1) the antioxidative properties of astaxanthin protect against oxidative stress in neurological diseases by activating the signaling pathway of extracellular signal-regulated protein kinase and upregulating the expression of Nrf2-regulated enzymes [76]; and (2) the anti-inflammatory properties of astaxanthin allow it to inhibit inflammatory molecule expression by suppressing the degradation of IκB-α and the translocation, or nuclear expression, of nuclear factor kappa B, thus lowering the expression level of proinflammatory cytokines [82]. The multiple neuroprotective properties of astaxanthin have been explored using various neurological disease models, indicating its potential ability to serve as a future neuroprotective candidate for treating chronic neurodegenerative disorders. However, it is still challenging to comprehensively evaluate the underlying mechanisms [36].(5)The antioxidant characteristics of astaxanthin have been applied in treating other human diseases, such as CVDs, alcoholic liver disease, and acute lung injury. CVDs accounted for approximately 16.7 million deaths worldwide in 2010. The financial burden for CVD prevention and treatment is speculated to reach up to USD 47 trillion on a global scale in the next 25 years [60]. Astaxanthin shows excellent potential for reducing atherosclerosis occurrence, but its potential must be further examined, as it has not been studied in humans [60]. Apart from CVDs, researchers have found that astaxanthin can reduce hepatic inflammation and lipid dysmetabolism in alcoholic liver diseases, providing new insight into a promising nutrition-related therapy for alcoholic liver disease [83]. Furthermore, astaxanthin treatment significantly decreased cecal ligation and puncture-induced lung damage, as well as the resulting mortality rate in rats [84]. The putative mechanisms underlying the protective effects against lung injury suppress inflammatory responses and inhibit NF-κBP65 expression [73]. Additionally, astaxanthin supplementation may help healthy people recover from the mental fatigue induced by oxidative stress [85]. Therefore, the antioxidant properties of astaxanthin have broad and promising applicability in inflammation-associated human diseases. Moreover, the combination of astaxanthin with PUFAs may shed new light on reducing inflammation-driven infections in humans [86].

### 2.4. Beta-Glucan

In recent years, the versatile phototrophic protist *Euglena gracilis* has emerged as a promising candidate for application-driven research and commercialization. It is an excellent source of dietary proteins, (pro)vitamins, lipids (DHA/EPA), and beta-glucan paramylon only found in euglenoids [87]. Based on this, paramylon is already marketed as an immunostimulatory agent in nutraceuticals. Beta-glucan from Euglena algae is a new immune-support ingredient, as the naturally occurring beta-glucan content is recognized explicitly by immune cell receptors [88]. Preclinical and clinical studies support the positive effects of this microalga and its beta-glucan content on multiple aspects of immunity [87]. With immunity as a primary global health concern due to COVID-19, the immune stimulant is rapidly garnering commercial interest regarding this unique algal species [89]. A previous study found that the performance of paramylon was more effective in animal feed than two commercially available beta-glucan products derived from yeast [90]. *E. gracilis* can accumulate large amounts of reserved polysaccharide paramylon, which can constitute over 80% (*w*/*w*) of the biomass dried to a constant weight without oxidation [91].

Nakashima and colleagues report that paramylon isolated from *E. gracilis* Z exerts an immunoregulatory role in protecting against influenza virus infection in mice [92]. Oral administration of paramylon results in the production of higher amount of IL-1β, IL-6, IL-12 (p70), IFN-γ, IL-10, and IFN-β and significantly lower virus titer in the lung, indicating that paramylon serves to alleviate symptoms of influenza virus infection [89]. *E. gracilis* paramylon treatment induces the production of NO, TNF-α, and IL-6 through activation of the NF-κB and mitogen-activated protein kinase (MAPK) signaling pathways and thus activates the immune system in murine RAW264.7 cells [88]. Laminarin-type β-(1→3)-glucan from *Sargassum henslowianum* functions in regulating the intestinal microbiota composition by stimulating the growth of species belonging to Enterobacteriaceae while depleting *Haemophilus parainfluenzae* and *Gemmiger formicilis* [93]. Supplementation with β-1,3-glucan from *E. gracilis* may reduce and prevent upper respiratory tract infection (URTI) symptoms in humans, including fewer sick days, URTI symptoms, URTI symptom days, URTI episodes, and lower global severity [94] (Table 4).

## 3. Discussion

Vaccine antigens and antibodies have been produced in many species, including mammals, bacteria, yeast, plants, and microalgae. However, most therapeutic protein drugs are expensive and not widespread due to downstream processing hurdles, such as purification, cold chain supply, and injection administration. The first expression system is bacteria that cannot express the most complex eukaryotic proteins due to a lack of eukaryotic posttranslational modifications necessary for proper folding, bioactivity, and important modifications, such as phosphorylation and glycosylation [5]. Fungal systems exhibit high growth rates and ease of scalability due to their ability to modify proteins posttranslationally. However, significant issues for this system include N-linked glycosylation, as well as inadequate secretion and proteolysis [5]. Other eukaryotic expression systems, such as insect or mammalian cells, are used to produce approximately 50% of all licensed recombinant proteins with the ability to confer posttranslational modifications correctly. Their main disadvantage is the difficulty in scaling up their cost-intensive production processes.

Plant systems became popular after attempts to overcome these high production costs and reduce pathogenic contamination risk. In the last few decades, attempts to produce edible vaccines in whole plants has appeared attractive, but remain challenging due to low production levels and costly purification processes for products with no oral application [5]. Microalgae are showing promise in recombinant protein research and may serve as a sustainable and cheap source for vaccines and antibodies in the future. Some advantages of microalgal expression systems include: (1) the ease of obtaining stably transformed lines with unique advantages over other plant systems, such as tobacco; (2) the rapid accumulation of biomass without seasonal constraints; (3) the genetic contamination of other crops is avoidable using enclosed bioreactors culture systems; (4) their GRAS classification by the US FDA; and (5) the ability to be quickly freeze-dried and an algal cell wall that assures an encapsulation effect [99]. Compared to conventional vaccine formulations, microalgae-based technology is becoming an ideal system, with the advantages of low costs and simpler logistics. Research on microalgal recombinant protein platforms is growing, focusing on investigating and developing microalgae-originated vaccines and antibodies.

### 3.1. Algae-Based Antibodies and Vaccines

This first algae-based antibody was transformed into the chloroplasts of *C. reinhardtii* [100]. Further application of antibody expression using chloroplasts from *C. reinhardtii* was based on its ability to fold complex proteins, including those with disulfide bond formations. One example is the malaria antigen. To detect malaria, the cell-traversal protein for ookinete and sporozoite antigens from *Plasmodium falciparum* was successfully expressed in the chloroplasts of *C. reinhardtii*. Accordingly, a highly sensitive and specific indirect enzyme-linked immunosorbent assay was developed and widely applied [101].

Moreover, Shahriari et al. reported the successful construction of anti-Newcastle disease virus recombinant subunit vaccines in the *C. reinhardtii* platform through an agrobacterium-mediated genetic transformation system. In detail, an agrobacterium-mediated genetic transformation was performed to express a chimeric gene construct that included hemagglutinin-neuraminidase and fusion epitopes of Newcastle disease virus in *C. reinhardtii* [102]. A recently developed expression system, Algevir, can efficiently express an antigenic protein (ZK) with a yield of up to 365 µg/g fresh weight [103]. Using the same algal species and the Algevir approach, another research group successfully carried out a highly efficient expression of a thermostable Alzheimer’s disease vaccine candidate (LTB: RAGE) [104]. The first example of an oral therapeutic produced from *N. oculate* is the antimicrobial peptide bovine lactoferricin, which can increase the survival rate fish against diseases by 85% [105]. In another study, after using a low expression level (2 ng/mg) of oral vaccination, a white spot syndrome virus (WSSV) subunit vaccine generated from *D. salina*, crayfish exhibited increased survival rates against WSSV infection [106]. Additionally, the hepatitis B virus surface antigen was produced using chloroplasts from the microalga *D. salina* but further oral applications have yet to be confirmed [107]. The diatom *P. tricornutum* had a successful fully monoclonal antibody created against the hepatitis B virus with a yield of 9% of total soluble protein [5,108].

### 3.2. Current Commercialization Hurdles

Algae have US FDA GRAS status and a new economic oral vaccination platform using green fluorescent proteins expressed in the chloroplasts of algal cells has been evaluated [109]. Thus, microalgae have the potential to become a drug delivery platform with the advantages of injection-free administration and none of the associated safety hazards [110], as well as comparatively simple requirements for cold transportation [109].

Although no clinical trials for algae-based vaccines are currently in progress, related technical challenges that need to be addressed include low protein production in the nuclear genome or no glycosylation in the chloroplast expression of papillomavirus, hepatitis B virus, and foot and mouth disease virus [111]. Moreover, microalgal cells can bio-encapsulate recombinant proteins due to their rigid cell wall, which could solve the degradability issue [109]. Additionally, controlled facilities for culturing transgenic algae avoid the potential flow of transgenes to an environment that could cause ecosystem disaster [109]. Although a few vaccines from microalgae have been applied in animal models, approval for their commercial production and clinical trials are still pending. Therefore, much work remains to be conducted for the large-scale commercial production of vaccine-related pharmaceuticals [112].

Several obstacles and disadvantages regarding drug delivery platforms need to be addressed (Table 5). For example, new technology, like nanoparticles, could solve the problem of mucosal tolerance and increase antigen bioavailability. However, no US FDA-approved compounds for edible vaccines are available [99]. Additionally, the majority of injectable vaccines are only available in Europe and North American for high-value fish species, such as salmon, trout, and sea bass [113].

Production of beta-glucan, fat-soluble vitamins, and sulfated polysaccharide enables the application of algae for human health or the medical field. Increases in glucan content can be achieved through optimized growth conditions. For instance, nitrate starvation enhanced the β-glucan content of *S. ovalternus* SAG 52.80 from 23.3% to 36.7% and of *Scenedesmus obtusiusculus* A 189 from 16% to 23%, while increased irradiance enhanced the β-glucan content of *S. obtusiusculus* A 189 from 6.4% to 19.5% [91]. Algal biomass and bioproduct content are critical and careful considerations and need to be considered to keep the balance. Although heterotrophic (HT) cultivation derived four times more algal cell dry weight than photoautotrophic (PT) cultivation [114], PT cultivation leads to a production of more antioxidants than HT and mixotrophic cultivation, but similar free amino acid in *Euglena gracilis* [115]. In addition, industrial-scale heterotrophic cultivation of *Eu. gracilis* is limited by the high cost of medium containing an organic carbon source such as glucose, vitamins (especially B1 and B12), medium sterilization, and cultivation environments to avoid contaminants [87]. A complex medium based on potato liquor from waste streams of the potato starch industry might be a cheaper alternative [116]. Development of biotechnological techniques, i.e., genetic modification or metabolic engineering of algal strains, shows great promise for enhanced algal biomass/bioproduct yields, although considerations need to be taken, as discussed in [116].

We have also noticed the pharmacokinetic issue of bioactive compounds derived from algae. Bioactive compounds such as beta-glucan (paramylon), fucoidan and griffithsin have a high molecular weight (>500), which means these compounds do not obey both Lipinski’s and Veber’s rules of drug-likeness and do not possess high degrees of predicted bioavailability. In other words, they do not have drug-likeness, it is very hard to make them into drugs, or they have a low absorption. However, astaxanthin can pass through the BBB, and a few of algae-derived molecules can do this [117].

## 4. Current Technological Difficulties of Microalgal Platforms

### 4.1. Genetic and Metabolic Engineering

Engineered microalgae should not affect the algal GRAS status due to genetic manipulation, as the clustered regularly interspaced short palindromic repeats (CRISPR)/Cas9 system will not introduce foreign DNA into the microalgal genome [4]. Using the (CRISPR)/Cas9 RNA-guided DNA cleavage defense-based gene knockout technique, Nymark et al. successfully constructed a CpSRP54 knockout mutant in the marine diatom *P. tricornutum* [119]. It was reported as the first CpSRP54 mutation in microalgae, the product of which is a member of the chloroplast signal recognition particle pathway. However, only a minimal CRISPR/Cas9 strategy has been carried out in microalgae compared to archaea and bacteria. As Cas9 nuclease production appears to be toxic for some microalgae, the generation efficiency of genome-modified strains is therefore limited. Correspondingly, a new strategy was developed using the same microalgae that avoided toxicity [120]. Metabolic engineering is also a powerful tool used to generate desired species for valuable compound production, such as pharmaceuticals. However, this technique is still in its infancy due to a lack of metabolic pathways [7]. Researchers have developed systems through biological and synthetic biological approaches to build effective photosynthetic microalgal cell factories to address such challenges. The rapid development of these innovative technologies will offer further chances for producing new active pharmaceutical ingredients [4].

### 4.2. Harvesting and Purification

Microalgal harvesting systems are quite expensive, which is a limiting step that must be overcome to preserve algal integrity and molecular stability. Techno-economic studies suggest that costs of microalgae harvesting account for 20–30% of the total production cost, while the extraction and purification of the products are critical processes that contribute up to 60% of the total production cost [121]. Research has focused on developing methods for the downstream processing of algal biomass, oil extraction, and biofuel production. However, solvent extraction can result in high toxicity if used for animal or human consumption. The CO_2_-based supercritical fluid extraction technique is non-toxic and can be applied to separate the lipids and other high-value compounds from the matrix in algae (dried biomass or disrupted concentrate), but the operational cost is high [122]. There are many challenges facing microalgal platforms, such as protein solubility [123], protein instability, aggregation in the body [124,125], posttranslational modifications related to biological function [9], and both cost and logistical considerations [111].

### 4.3. Scale-Up Technology

Currently, the scientific community lacks successful scaled-up examples (transfer from lab scale to industrially relevant). For instance, growth conditions for scaled-up microalgal recombinant protein production have not yet been established. Many of these plant-derived compounds can be produced by chemical synthesis, however, their overall yields are limited due to structure complexity, which may require difficult multistep regio- and stereo-specific reactions [122]. A potential robust platform for the commercial production of proteins is the application of chloroplasts, but it is limited due to no secretion of chloroplast-expressed proteins. Thus, cultivated microalgal cells must be processed prior to further product recovery and purification processes. Therefore, high purity standards can be met through harvesting, cell disruption (for intracellular compounds), and purification in order to recover high-value products. Compared to bacteria, yeast, or other microbial systems, further development and optimization procedures are needed to achieve comparable secretion yields of recombinant proteins in microalgae [122].

Natural proteins purified directly from human cell cultures or recombinant proteins are used in clinics worldwide, such as in human interferons produced by *Escherichia coli*. However, pharmaceutical production from human cell cultures is costly and purification is time-consuming [126]. Microalgal pharmaceutical production, based on bioreactor cultivation, may offer considerable advantages, especially for *C. reinhardtii*. However, further improvements are forthcoming. Currently, there are no microalgae-produced pharmaceuticals with regulatory approval for commercial use or clinical trials. Still, a few microalgae-derived recombinant proteins have been produced efficiently in cells (Table 6).

Regarding commercial applications, microalgae-produced recombinant proteins remain in the early stages of development. Compared to well-established expression systems for bacteria, yeast, plants, and human cells, enhancing the productivity and secretion efficiency is needed. Additionally, the time to produce personalized drugs from algae needs to be shortened to days instead of months. Despite their potential, many hurdles need to be overcome for the commercial production of microalgal pharmaceuticals, and extensive studies regarding their effects on human health should be conducted [7]. Clearly, much effort is still required to make the commercial large-scale production of microalgal factories for pharmaceuticals possible [112].

## 5. Conclusions

Microalgae have long been known as promising platforms due to their feasibility in modern manufacturing facilities, only using carbon dioxide, water, and sunlight to produce bioactive compounds. However, from a sustainability perspective, certain obstacles remain to be overcome for the use of this microalgal production system. Secondary metabolites, such as antioxidant, antiviral, antifungal, antitumor, antimalarial, and other bioactive compounds, have industrial potential. However, it remains a significant challenge that microalgae compete with current well-established platforms, partially due to the lack of secretion of recombinant proteins into the media. Therefore, the potential of microalgae as cell factories has yet to be realized, and future efforts should concentrate on exploiting their distinctive advantages, such as their GRAS status, inexpensive culturing, and possible scalability. Future microalgal development work will need to focus on the following: (1) the safe evaluations of transgenic strains; (2) the development of recombinant protein standards executed under Good Manufacturing Practices conditions; (3) preclinical trials; and (4) complete clinical trials. Despite these limitations and challenges, the future is promising for algae-based pharmaceuticals.

## Figures and Tables

**Figure 1 marinedrugs-19-00703-f001:**
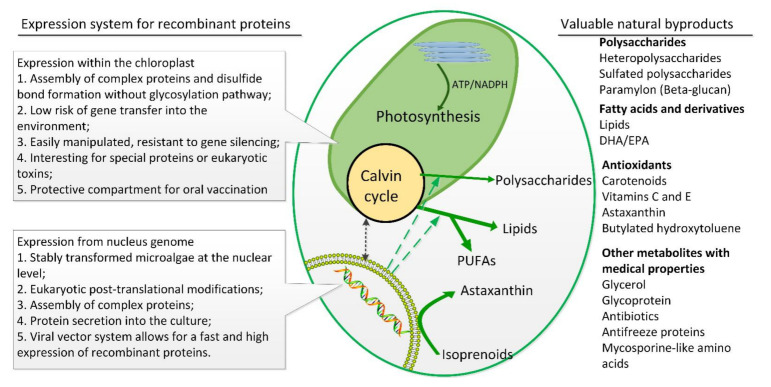
Schematic overview of microalgal cells as solar-powered factories to produce pharmaceuticals: recombinant proteins and valuable natural byproducts with medical properties. The main advantages of microalgae as cell factories are being fueled by photosynthesis, carbon dioxide-neutral, rapid growth rates, robust, low-cost cultivation, easily scalable, no risk of human pathogenic contamination, and valuable natural byproducts.

**Figure 2 marinedrugs-19-00703-f002:**
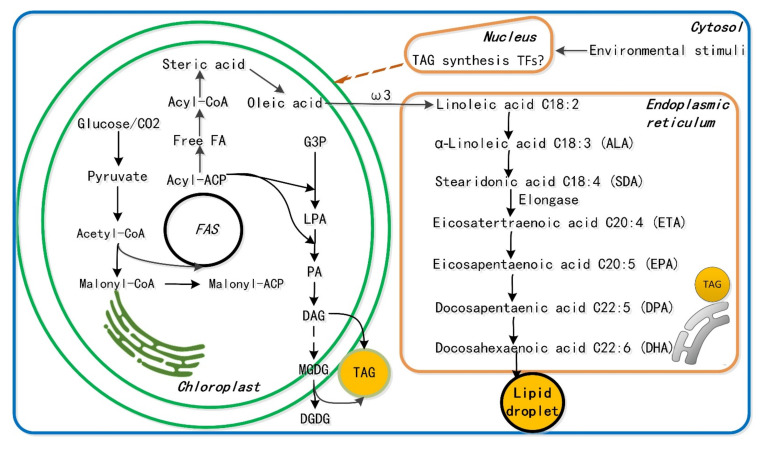
Schematic overview of fatty acid biosynthesis, allowing us to propose hypothetical routes for unknown pathways such as of PUFA biosynthesis. Remarkably, microalgae are the only form of life which can readily produce PUFAs by directly using the Sun’s energy. FA, fatty acid; FAS, fatty acid synthase; TFs, transcript factors; G3P, glycerol-3-phosphate; DAG, diacylglycerol; LPA, lysophosphatidic acid; TAG, triacylglycerol.

**Figure 3 marinedrugs-19-00703-f003:**
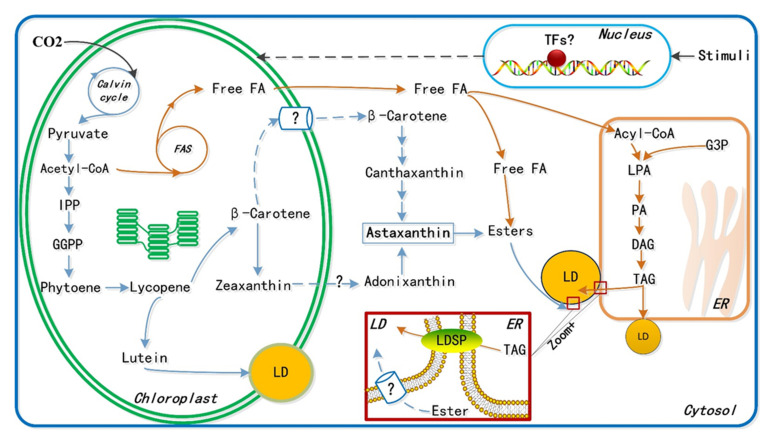
Overview of the proposed central biosynthesis pathways of astaxanthin (blue arrow) and lipid (orange arrow). Question marks represent the unresolved steps in pathways or proteins in membrane which will be primary targets to be identified via omics analysis. IPP, isopentenyl pyrophosphate; GGPP, geranylgeranyl pyrophosphate; FA, fatty acid; FAS, fatty acid synthase; TFs, transcript factors; G3P, glycerol-3-phosphate; DAG, diacylglycerol; LD, lipid droplet; LPA, lysophosphatidic acid; TAG, triacylglycerol; LDSP, lipid droplet surface protein; ER, endoplasmic reticulum.

**Table 1 marinedrugs-19-00703-t001:** Eicosapentaenoic acid (EPA, 20:5n-3) of algae/other marine organisms.

Name	Algae/Other	Study Model	Positive/Negative Control	Dose/Concentration Range	Effect		Ref.
					ABTS^+^ (µg/mL), IC_50_	DPPH (µg/mL), IC_20_	
Algae	*Chlorella vulgaris*	ABTS^+^, DPPH	control lipid: replace ABTS^+^ with ethanol	25, 125, 250, 500 µg/mL	51.1	50.5	[12]
	*Chlorococcum amblystomatis*				52.6	58.4	
	*Scenedesmus obliquus*				29.4	89.1	
	*Tetraselmis chui*				40.9	225.7	
	*Phaeodactylum tricornutum*				57.3	75.4	
	*Spirulina* sp.				38.7	96.6	
	*Nannochloropsis oceanica*				101.9	175.6	
	*Palmaria palmata*	ABTS^+^, DPPH	control lipid: replace ABTS^+^ with ethanol	25, 50, 100, 250 µg/mL	26.2	IC_30_ = 171	[13]
	*Grateloupia turuturu* Yamada	ABTS^+^, DPPH; COX-2	control lipid: replace ABTS^+^ with ethanol	12.5, 250 µg/mL	130.4	IC_50_ = 130.4	[14]
					33 *		
Sea urchin	*Glyptocidaris crenularis*	-	-	11.4% of DW	-		[15]
	*Strongylocentrotus intermedius*	-	-	20.0% of DW	-		
	*Strongylocentrotus nudus*	-	-	9.3% of DW	-		
Sea urchin	*Strongylocentrotus droebachiensis*	human mononuclear U937 cell	*Escherichia coli* LPS	15.2 µg/mg, 9.6% of DW	anti-inflammatory activity: 88% of MAPK p38 inhibition at a dose of 0.033 µg/mL, COX-1, and COX-2.		[16]

Antioxidant activity: ABTS^+^ and DPPH. Anti-inflammatory activity: COX-2. * COX-2 IC50ca.

**Table 2 marinedrugs-19-00703-t002:** Sulfated polysaccharides from algae.

Name	Algae	Monosaccharide Composition	Molecular Weight	Study Model	Positive/Negative Control	Dose/Concentration Range	Effect	Ref.
Fucoidan (sulfate content 27.0%)	*Fucus vesiculosus*	fucose 73.5 mol%, galactose 3.7 mol%, glucose 11.8 mol%, xylose 6.6 mol%, mannose 0.2 mol%, arabinose 0.2 mol%	735 kDa ultrasound-assisted extraction	human mononuclear U937 cells	*Escherichia coli* LPS	10 mg/mL	inhibit hyaluronidase and DPP-IV. IC_50_ for DPPH 35 μg/mL, AA 0.32, BHA 0.59 μg/mL	[32]
				human DPP-IV	sitagliptin	0.2–200 μg/mL		
				platelet-poor plasma	-	1.6, 3.2, 4.8, 6.3, 9.1, 10 μg/mL; 80, 310, 380, 450 μg/mL		
Fucoidan (sulfate content 27.1%)	*Ascophyllum nodosum*	fucose 41.2 mol%, galactose 6 mol%, glucose 6 mol%, xylose 15 mol%, mannose 11.3 mol%, uronic acid 24.6 mol%	34.4 kDa	-	-	-	30.4% scavenging of DPPH at 10 mg/mL	[33]
Sulfated polysaccharides (F1, F2)	*Corallina officinalis*	galactose, xylose	-	-	-	-	-	[34]
Crude fucoidan (sulfate content 22.8%)	*Undaria pinnatifida*	fucose 29 mol%, xylose 30 mol%, galactose 23 mol%, glucose 3 mol%, uronic acid 4 mol%	>300 kDa	-	-	AA 244 μg/mL, BHA 235 μg/mL, Trolox equivalent	7.4 μg/mL Trolox equivalent	[35]
Purified fucoidan (sulfate content 20.0%)		fucose 27 mol%, xylose 3 mol%, galactose 18 mol%, glucose 2 mol%, uronic acid 4.6 mol%	300 kDa	-	-		9.0 μg/mL Trolox equivalent	
Standard fucoidan (sulfate content 30.0%)		-	lower than 10 kDa				8.8 μg/mL Trolox equivalent	
Sulfated α-L-fucan	9072-19-9, Cayman Chemical, Ann Arbor, MI, USA		unknown	oral pathogens: *Candida albicans* (JCM1537), *Streptococcus mutans* (JCM5705), *Porphyromonas gingivalis* (JCM8525)	antibiotic (positive control), PBS (negative control). Endotoxin-neutralizing of LPS	100 mg/mL	strong antimicrobial activity, crude fucoidan showed stronger inhibition effect than purified fucoidan	[36]
Fucoidan extract with a low molecular weight prepared by glycosidase digestion (sulfation 14.5%)	*Cladosiphon novae-caledoniae*	fucose (73%), xylose (12%), and mannose (7%)	digested low-molecular-weight fraction (72% <500 kDa) and non-digested fraction (>28%, 800 kDa peak)					
Crude fucoidan sulfate (23%)	*Fucus vesiculosus*	fucose (33%), uronic acid (8%),	20–200 kDa					
Purified (>95%) fucoidan	Fucus vesiculosus	-	68.6 kDa					
Fucoidan	Durvillaea antarctica	mole ratio: 1.1 fucose, 26.2 glucose, 0.9 xylose, 2.9 mannose, 2.7 sorbose	482 kDa	-	-	-	-	[37]
	*Sarcodia ceylonensis*	mole ratio: 5.3 glucose, 1.2 arabinose, 14.4 mannose, 2.8 sorbose	466 kDa	-	-	-	-	
	*Ulva lactuca* L.	mole ratio: 0.2 fucose, 1.9 glucose, 0.2 galactose, 0.5 arabinose, 0.3 xylose, 6.7 mannose, 0.5 sorbose	404 kDa	-	-	-	-	
	*Gracilaria lemaneiformis*	mole ratio: 4.5 glucose, 1.8 xylose, 18.8 galactose, 6.0 frucose	591 kDa	-	-	-	-	
	macroalga *Ulva lactuca*	molar rate 1.1, 1.9, 0.2, 0.5, 0.3, 6.7, and 0.5 for fucose, glucose, galactose, arabinose, xylose, mannose, and sorbose	466 kDa	-	-	-	-	[38]
Fucoidan (LJSF4) (sulfate 30.7%, yield 7.5%)	*Saccharina-japonica*	fucose, galactose, rhamnose, xylose, mannose	104.3 kDa	zebrafish	LPS-induced toxicity	12.5–50 μg/mL	reduces the cell death rate, decreases the production of nitric oxide, ROS and cytokines, including TNF-α, IL-1β, and IL-6. LJSF4 pre-treatment significantly decreased the heart rates of zebrafish larvae and even reduced to 103.2% at 50 μg/mL	[39]
Fucoidan (Fu)	*Fucus vesiculosus*	-	-	human pulmonary microvascular endothelial (HPMEC-ST1.6R) cells/chick chorioallantoic membrane	-	-	enhanced viability of endothelial cells and vascularization	[40]
Sulfated polysaccharide (PS)	*Turbinaria ornata*	glucopyranose, fucopyranose	unknown	rat	LPS, LPS + dexamethasone, LPS + PS/normal control	2.5, 5, 10 mg/kg body weight	prevents LPS-induced systemic inflammation in the cardiac tissue, PS mitigates inflammation by repressing and/or inhibiting iNOS, NFκB, and PI3K/Akt pathway	[41]
Sulfated polysaccharide (PS)	*Turbinaria ornata*	-	unknown	RAW 264.7 macrophages	DMEM medium, LPS	10, 20, 40 μg/mL	increases the antioxidants GSH and SOD, significantly reduces mRNA levels of IL6 and TNFα	[42]
SBPs (sulfate, 24.1%)	*Sargassum binderi*	fucose, galactose, glucose, mannose, arabinose, rhamnose	average 2.867 × 10^5^ g/mol for SBP-fraction 4	macrophages (RAW 264.7)/zebrafish	LPS + SBPs/control (neither LPS nor SBP)	25, 50, 100, 200 µg/mL	inhibits COX-2 and iNOS protein levels in LPS-activated macrophages and reduces cell death and NO production in LPS-treated zebrafish larvae	[43]
lambda-carrageenan (λ-CGN), commercial	-	α-galactose	-	influenza A and B viruses, d severe respiratory syndrome coronavirus 2 (SARS-CoV-2)/mice	virus infected MDCK or Vero cells/mock-infected; λ-CGN/p-KG03 or EGCG	10, 100, 300 µg/mL	targets viral attachment to cell surface receptors and prevents virus entry	
acidic polysaccharide of the Coccomyxa gloeobotrydiformis Nikken strain (AEX)	*Coccomyxa gloeobotrydiformis*	galactose, mannose, glucose, arabinose, xylose, rhamnose	-	MDCK cells; human influenza A virus,	MDCK cells inoculated with human influenza A virus/uninfected living cells	26–70 µg/mL	prevents the cell attachment and/or penetration of influenza virus; prevents the interaction of virus and host cells	[44]
acidic polysaccharide of the Coccomyxa gloeobotrydiformis Nikken strain (AEX), Nikken Sohonsha Corporation (Hashima, Gifu, Japan)	*Coccomyxa gloeobotrydiformis* Nikken strain	galactose, mannose, glucose, arabinose, xylose, rhamnose	-	chicken immune cells; IBDV	Vero cells incubate with IBDV Ts strain and AEX, IBDV live vaccine; Con A as positive control	12.5, 25, 50, 100 mg/mL	represses IBDV replication by the deactivation of viral particles or by interfering with adsorption in vitro, and reduces the IBDV viral titer in the chicken bursa of Fabricius	
GFP (sulfate, 19.9%)	*Grateloupia filicina* (19.7% yield)	molar ratio: 0.01 Man, 0.02 Glc A, 0.07 Glc, 1 Gal, 0.1 Xyl, 0.05 Fuc	unknown	avian influenza virus (AIV)/MDCK cells, mice	MDCK cells in DMEM as a control, MDCK cells in sulfated polysaccharides dissolved in DMEM	50, 100, 500 µg/mL	stimulation of IFN-γ production, IL-4 stimulation	[45]
UPP (sulfate, 13.5%)	*Ulva pertusa* (12.1% yield)	molar ratio: 0.06 Man, 1 Rha, 0.53 Glc A, 0.19 Glc, 0.09 Gal, 0.39 Xyl, 0.02 Fuc	-			-		
SQP (sulfate, 5.6%)	*Sargassum qingdaoense* (7.1% yield)	molar ratio: 0.56 man, 0.13 Glc A, 0.37 Glc, 0.6 Gal, 1 Fuc						
(Sulfate, 31.0%) (1→4)-linked β-L-arabinopyranose	*Enteromorpha* *clathrata*	arabinose (80.5%), rhamnose (10.7%), galactose (4.8%), glucuronic acid (4.0%)	511 kDa	human plasma samples	heparin as a reference	10, 20, 50, 100 µg/mL	stimulates TNF-α expression in serum and induces lymphocyte proliferation	[46]
Fucan SV1 (sulfate 22.6%)	*Sargassum horneri*	fucose 36.8%, galactose 17.1%, xylose 8.1%, glucuronic acid 11.1%, mannose 12.4%	unknown	rat, RAW 264.7 (mouse leukemic monocyte macrophage cell line)	DMEM medium	0.3–2.5 mg/mL	reduces edema and cellular infiltration	[47]
Glucan	*Sargassum horneri*	T-D-Glcp, 1,3-D-Glcp, 1,6-D-Glcp and 1,3,6-D-Glcp	578 kDa	human colon cancer DLD cells	-	-	inhibits human colon cancer DLD cell growth	[48]
Extracted sulfated carrageenan (ESC)	*Laurencia papillosa*	ι-carrageenan	-	MDA-MB-231 cancer cell line	-	50 µM	inhibits breast cancer cells (MDA-MB-231) via apoptosis regulatory genes	[49]
p-KG103	*Gyrodinium impudium*	-	-	mice	-	100 or 200 mg/kg body weight	activates NO production to stimulate the production of cytokines and prevent tumor cell growth	[50]

Antioxidants (SOD, GSH NO, and LPO), pro- and/or anti-inflammatory markers (IL6, IL10, TNFα, and iNOS), peripheral blood molecular cells (PBMCs), infectious bursal disease virus (IBDV), tumor necrosis factor α (TNFα), Madin–Darby canine kidney (MDCK) cells, phosphate-buffered saline (PBS), dipeptidyl peptidase-IV (DPP-IV), glutathione (GSH), malondialdehyde (MDA/LPO), nitric oxide (NO), superoxide dismutase (SOD).

**Table 3 marinedrugs-19-00703-t003:** Summary of recent studies of astaxanthin application in cerebrovascular disease and significant findings.

Disease Type	Model	Effect	Significant Findings	Reference
Alzheimer’s disease	primary porcine brain capillary endothelial cells (pBCEC), and in 3xTg AD mice	neuroprotective effect	astaxanthin reduces BACE-1 (activity) and Aβ/oligomers in mBCEC and deeper regions of the brain, affecting not only the clearance but also the generation of Aβ	[63]
	primary hippocampal neurons	neuroprotective effect	astaxanthin protects neurons from the harmful effects of A𝛽Os on mitochondrial ROS production, NFATc4 activation, and RyR2 gene expression downregulation	[64]
	adult hippocampal neurogenesis (AHN) and spatial memory using a mouse model	neuroprotective effect	novel insights into the neurogenic effect of astaxanthin on hippocampus-dependent cognitive function by preventing cognitive impairment	[65]
Parkinson’s disease	an aged mouse model	multiple biological activities	astaxanthin attenuates neurotoxicity of Parkinson’s disease in both young and aged mice	[66]
Subarachnoid hemorrhage injury	adult male Sprague Dawley rats	anti-inflammation	astaxanthin increases sirtuin one levels and inhibits the TLR4 signaling pathway, then reduces the proinflammatory response and second brain injury	[67]
	adult male SD rats	neuroprotective effect	astaxanthin attenuated SAH-induced cerebral vasospasm and reduced neuronal apoptosis	[68]
	adult male SD rats, prechiasmatic cistern SAH model	antineuroinflammation	astaxanthin shows neuroprotective effect with the possible mechanism of suppression of cerebral inflammation	[69]
	male Sprague Dawley rats, prechiasmatic cistern SAH model	neurovascular protection	astaxanthin reduces the expression and activity of MMP-9 and ameliorates brain edema, BBB impairment, neurological deficits, and TUNEL-positive cells	[70]
	adult male SD rats	neuroprotective effects	astaxanthin attenuates SAH-induced EBI by enhancing neuronal survival and mitochondrial function	[68]
	male ICR mice	antioxidant activity	astaxanthin can suppress learning and memory impairment and attenuate oxidative stress	[71]
	astrocytes isolated from the cerebral cortices of neonatal C57BL/6 mouse pups	anti-inflammatory, neuroprotective	astaxanthin inhibits NKCC1 expression and reduces the expression of NF-κB-mediated proinflammatory factors	[72]
Acute cerebral infarction (stroke)	male C57BL/6 mice	anti-inflammatory, neuroprotective	astaxanthin ameliorates AQP4/NKCC1-mediated cerebral edema and then reduces TBI-related injury in brain tissue	[73]
	stroke-prone spontaneously hypertensive rats	antithrombotic, antihypertensive	antihypertensive and antithrombotic properties of astaxanthin	[74]
	male Sprague Dawley rats	neuroprotective effect	astaxanthin ameliorates ACI via the suppression of oxidative stress and upregulation of BDNF and NGF mRNA	[75]
Spinal cord injury	adult male Wistar rats	antineuroinflammation	Astaxanthin inhibits glutamate-initiated signaling pathway and inflammatory reactions in the secondary phase of SCI	[76]
	adult male Wistar rats	anti-inflammatory, neuroprotective	astaxanthin can reduce neuronal apoptosis and improves functional recovery after SCI	[77]
Traumatic brain injury	male adult ICR mice	neuroprotective action	astaxanthin reduces cortical lesion volume, neuronal cell loss, and neurodegeneration in the cortex by simulating neurotrophic factors and promoting synaptic survival	[78]
Cognitive disease	adult male Sprague Dawley rats, amygdala kindling, epilepsy	neuroprotective effects	astaxanthin attenuates oxidative damage and lipid peroxidation and inhibits the mitochondrion-related apoptotic pathway	[79]
	male C57BL/6J	neurodegenerative disease	astaxanthin modulates cognitive function and synaptic plasticity	[80]
	male Wistar rats	antioxidant	astaxanthin inhibits oxidative stress and inflammatory responses	[59]
Peripheral vascular disease	human umbilical vein endothelial cells (HuVecs)	antioxidant	astaxanthin inhibits Hcy-induced endothelial dysfunction via the suppression of Hcy-induced activation of the VEGF-VeGFr2-FaK signaling axis	[68]

**Table 4 marinedrugs-19-00703-t004:** Effect of beta-glucan.

Name	Algae	Effect	Host	Ref.
β-glucan	*Euglena gracilis*	alleviates diarrhea of F18 *E. coli*-infected pigs by enhancing gut integrity, stimulates T cell activation, and reduces inflammation	pig	[95]
Paramylon (storage β-1,3-glucan)	*Eu. gracilis*	directly stimulates intestinal epithelial cells via Ca^2+^ signaling, stimulates dendritic cells (DCs) in Peyer’s patches	mice	[89]
β-1,3-glucan	*Eu. gracilis* Kelbs var. bacillaris ATCC PTA-123017 strain	reduces and prevents upper respiratory tract infection (URTI) symptoms in humans including incidence, duration, and severity: fewer sick days, URTI symptoms, URTI symptom days, URTI episodes, and lower global severity	humans	[94]
Paramylon	*Eu. gracilis* Z	higher survival rates from influenza virus infection, significantly lower virus titer in the lung, increased inflammatory cytokines (higher amount of IL-1β, IL-6, IL-12 (p70), IFN-γ, IL-10). Induces CD8+ T cells and/or NK cells	mice	[92]
β-1,3-glucan	*Eu. gracilis*	upregulates inducible nitric oxide synthase (iNOS) and increases secretion of nitric oxide (NO), interleukin (IL)-6, and tumor necrosis factor (TNF)-α, activates the nuclear factor-κB (NF-κB) and mitogen-activated protein kinase (MAPK) signaling pathways	murine RAW264.7 macrophages	[88]
Laminarin-type β-(1→3)-glucan	*Sargassum henslowianum*	regulates the intestinal microbiota composition by stimulating the growth of species belonging to *Enterobacteriaceae* while depleting *Haemophilus parainfluenzae* and *Gemmiger formicilis*	gut microbiota via in vitro fermentation with human fecal cultures	[96]
β-glucans (commercial)	2 g/d Aleta™, Kemin Industries, Inc., Des Moines, IA, USA	promotes a higher abundance of *Alloprevotella* and *Holdemanella;* beneficial to fecal bacteriome and consequently to the health and performance of dairy calves (affects the fecal bacterial community with possible consequences on animal growth and health)	newborn Holstein calves	[97]
β-glucan nanoparticles (β-GluNPs)	*Gracilaria corticata*	Fabrication of the water-soluble β-GluNPs using β-glucan. β-GluNPs have potential antibacterial activity against Gram-positive bacterial isolates. Stronger cytotoxicity efficacies of β-GluNPs than free β-Glu for breast cancer cells	bacterial *Staphylococcus aureus* MTCC 96, *Bacillus subtilis* MTCC-2387, *Pseudomonas aeruginosa* MTCC 424, *Proteus vulgaris* MTCC 426human, MCF-7 breast cancer cell line	[98]

**Table 5 marinedrugs-19-00703-t005:** Comparison among different systems for the production of therapeutic proteins.

Properties	Bacteria	Yeast	Plant	Insect	Human Cell Line	Algae
Production cost	inexpensive	inexpensive	inexpensive (€0.0045/g)	expensive	expensive (€70–140/g)	inexpensive (€0.0022/g)
Human pathogen	susceptible	non-susceptible	non-susceptible	susceptible		non-susceptible
Contamination	high risk	high risk	moderate risk	low risk	high risk	low risk
Glycosylation	nonglycosylation	glycosylation	nonglycosylation	non-human glycosylation	glycosylation	nonglycosylation
Therapeutic efficacy	low	good	high	good	good	high
Half-life	n/a	low	high	n/a	low	high
Production time	a few weeks to a month	short	a few months to years	short	a few months to a year	a few weeks to months
Current market	32%	15%	less than 10%	less than 10%	43%	less than 10%

Note: information sourced from [118].

**Table 6 marinedrugs-19-00703-t006:** Summary of recent studies of the vaccine production system in microalgae.

Product	Target Disease	Microalgae	Vector/Transformation	Expression Level	Reference
Antigen	Newcastle disease virus	*Chlamydomonas reinhardtii*	pGH vector/Agrobacterium	N/A	[102]
Oral vaccine	malaria	*Chlamydomonas reinhardtii*	SapI/HindIII pASapI vector/chloroplast	1.5% TSP	[101]
Vaccine	Alzheimer’s disease	*Schizochytrium* sp.	Algevir system/nuclear	380 µg LTB:RAGE/g	[104]
	bursal disease virus	*Chlorella pyrenoidosa*	pART27 binary vector/Agrobacterium	N/A	[127]
Antigen	Zika virus (ZIKV)	*Schizochytrium* sp.	Algevir	365 μg g^−1^	[128]

## Data Availability

Not applicable.

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
