# Peer review of "Recent Advancements and Future Perspectives of Microalgae-Derived Pharmaceuticals"

_marinedrugs, 2021, doi:10.3390/md19120703_

Round 1
Reviewer 1 Report
The authors of the manuscript examined an important aspect of microalgae. While reading this manuscript, I had some recommendations and questions.
- For section 2. Please specify which vaccines are produced by microalgae. Provide literary references. Specify which metabolites (primary or secondary) are vaccines and antibodies produced by microalgae.
- For section 2.1. Compare the PUFAs of microalgae with other marine organisms, such as sea urchins (for example, https://doi.org/10.1016/j.foodchem.2017.08.076, https://doi.org/10.3390/md15120365, etc.). Supplement the section for better visualization with a table indicating: name of the lipid component, source of microalgae, study model, positive / negative control, dose / concentration range, effect, literature reference. Discuss the effect of the type of algae for PUFAs extraction on its effectiveness.
- For section 2.2. Sulfated polysaccharides from microalgae are derived from xylose, glucose, and galactose according to your manuscript. Compare the activity of fucose-derived sulfated polysaccharides with your data. Discuss the key aspects in terms of molecular weight and the degree of sulfation of compounds extracted from microalgae. Compare with similar ones from algae (e.g. https://doi.org/10.1016/j.biopha.2009.03.005, https://doi.org/10.3390/md18050275, https://doi.org/10.1016/j.carbpol .2011.12.029). Supplement the section with a table indicating: name of sulfated polysaccharide, microalgae source, monosaccharide composition, molecular weight, study models, positive / negative control, dose / concentration range, effect, literature reference. Discuss the effect of the type of algae for the extraction of sulfated polysaccharide on its effectiveness.
- For section 2.3. Seaweed contains not only astaxanthin but other fat-soluble vitamins as well. Complete the section with relevant information. Supplement Table 1 with the dose / concentration studied, positive / negative controls. The effect of astaxanthin on cerebrovascular disease is important, but it has other effects in a variety of diseases. Perhaps this section should be modified into a separate manuscript, and this manuscript should consider the various fat-soluble vitamins.
- For section 2.4. Supplement this section with more detailed information on preclinical and especially clinical beta-glucan data.
- For section 2. Include an additional section on vaccines and sera from microalgae.
- For section 3. Supplement the corresponding section with the achievements and difficulties in the production of beta-glucan, fat-soluble vitamins, and sulfated polysaccharides.
- A key aspect of the active compounds is their bioavailability. Consider the problem of absorption of microalgae-derived pharmaceuticals (e.g. https://doi.org/10.3390/md18110557).
Author Response
For section 2. Please specify which vaccines are produced by microalgae. Provide literary references. Specify which metabolites (primary or secondary) are vaccines and antibodies produced by microalgae.
Response: We have removed the sentence because section 3 focuses on the vaccines.
For section 2.1. Compare the PUFAs of microalgae with other marine organisms, such as sea urchins (for example, https://doi.org/10.1016/j.foodchem.2017.08.076, https://doi.org/10.3390/md15120365, etc.). Supplement the section for better visualization with a table indicating: name of the lipid component, source of microalgae, study model, positive / negative control, dose / concentration range, effect, literature reference. Discuss the effect of the type of algae for PUFAs extraction on its effectiveness.
Response: Thanks a lot for the comments.
Interestingly, besides provide extraction of PUFAs, the addition of red algae extracts from Gracilaria chilensis, Gelidium chilense, Iridaea larga, Gigartina chamissoi, Gigartina skottsbergii and Gigartina radula preserves the degradation of almost a half of DHA and 25% of EPA upon cooking heat treatment, it also inhibits the development of diverse pathogenic bacteria (Bacillus cereus, Escherichia coli, Staphylococcus aureus, Pseudomonas aeruginosa, Proteus mirabilis, and Salmonella enteritidis) [1].
To produce high PUFA yields, different extraction techniques as well as various solvents and solvent systems have been evaluated. Pressurized liquid extraction (PLE, also known as accelerated solvent extraction) and microwave-assisted solvent extraction (MAE) technique were compared for PUFA particularly EPA extraction in Phaeodactylum tricornutum [2]. PLE method provides higher extraction yields although MAE technique results in extracts with higher antioxidant activity, which may be caused by the different optimized extraction temperature (50 ℃ and 30 ℃ for PLE and MAE, respectively) [2]. Moreover, EPA extraction was determined using tight, thick and complex multilayered structure microalgal Nannochloropsis oceanica. As a result, EPA content extracted from spray-dried biomass using different solvents system: dichloromethane/methanol (DM) ≈ chloroform/methanol (CM) > ethanol extraction assisted with ultrasound probe (USP) > ethanol ≈ dichloromethane/ethanol (DE) > ethanol extraction assisted with ultrasound bath [3]. DM, CM and ethanol with USP solvents extracted similar yields of EPA, with assistance of USP, ethanol extraction leads to about 35% increase of EPA content compared to ethanol alone [3].
For section 2.2. Sulfated polysaccharides from microalgae are derived from xylose, glucose, and galactose according to your manuscript. Compare the activity of fucose-derived sulfated polysaccharides with your data. Discuss the key aspects in terms of molecular weight and the degree of sulfation of compounds extracted from microalgae. Compare with similar ones from algae (e.g. https://doi.org/10.1016/j.biopha.2009.03.005, https://doi.org/10.3390/md18050275, https://doi.org/10.1016/j.carbpol .2011.12.029). Supplement the section with a table indicating: name of sulfated polysaccharide, microalgae source, monosaccharide composition, molecular weight, study models, positive / negative control, dose / concentration range, effect, literature reference. Discuss the effect of the type of algae for the extraction of sulfated polysaccharide on its effectiveness.
Response: Thanks a lot for the positive comments. The following discussions were provided:
Biological functions of sulfated polysaccharides is depend on the presence and spatial positioning of sulfo groups (sulfate content) [4], molecular weight, fucose content and polyphenols [5]. For polysaccharides, sulfated modification enable/potentiate its antioxidant activity, and the disappearance of antiviral bioactivities is highly associated with the removing of sulfated groups [6], the anticoagulant activity of full or partially desulfated polysaccharides is lost compare to the native polysaccharide [7]. For instance, 6-O-desulfonated, 1→3 linked polysacharides eg. curdlan and galactan, lost 50% of the anticoagulant activity [4]. Moreover, sulfated polysaccharides with higher molecular weight shows better anti-bioactivity. One example is the fully sulfated galactan with the smallest molecular weight showed the lowest antithrombin activity compared to xylan, amylose, cellulose and curdlan [4]. With similar 27% sulfate content, 735 KDa of fucoidan from seaweed Fucus vesiculosus results in IC50 at 0.35 μg/mL [5], where 34.4 KDa of fucoidan from seaweed Ascophyllum nodosum causes 30.4% scavenging of DPPH at 10 mg/mL [8] (a lower IC50 value corresponds to a stronger antioxidant activity of the sample). However, there is also controversy report that low molecular weight polysaccharides LBP from Lilii Bulbus indicates better immunomodulatory effects than those with higher molecular weight [9]. λ-carrageenan from Chondrus ocellatus with molecular weights of 9.3 and 15 KDa showed better antitumor and immunomodulation activities [10].
Fucoidans is valuable to keep oral hygiene and dental caries prevention. Fucoidans show strong antimicrobial activity against Gram-negative bacteria Porphyromonas gingivalis as well as gram-positive bacteria Streptococcus mutans and Candida albicans, either through specific binding to pathogens that can directly neutralize endotoxin or using unknown mechanism [11]. Moreover, crude fucoidans indicating stronger inhibitory effect on adhesion to teeth than pure fucoidans [11].
Extraction method affects the yields of sulfate polysaccharide compounds and antioxidant properties [12, 13]. He and colleges evaluate different technologies for extraction of sulfate polysaccharide including microwave-assisted extract (MAE), hot-water extract, ultrasound-assisted extract. With the same ratio of water to raw material, MAE extracts had the highest polysaccharide yield (9.6%), while the hot-water extracts caused the lowest content of polysaccharide yield (8.5%) for Gracilaria lemaneiformis [12]. MAE may offer a rapid delivery of energy to the solvent and thus microwave radiation can be focused directly onto the sample, therefore, the heating is more efficient with shorter processing time (20 min vs 60 min for hot-water extract) [12].
Extraction temperature plays an important role on sulfate polysaccharide extraction, the molecular weight, sulfate content and antioxidant activities of extracted fucoid varied with extraction temperature using MAE method [8], probably because of the difference in produced monosaccharide composition: fucose at 90 ℃ and fucoidan at 150 ℃ for Ascophyllum nodosum [8]. The highest fucoidan yield (16.1%) was obtained using MAE with 120 ℃ for 15 min [8]. Although the yield of fucoidan was 21.0% using the conventional hotplate heating for 9 h, balance between yield and extraction time need to be considered [8]. In addition, pressure used for MAE extraction strongly influence the fucoidan compositions: galactose contents increased with the increase of pressure from 30 psi to 120 psi, fucose was only present at 30 psi while xylose was only present at 120 psi condition for seaweed [14]. Technique is developing towards increasing the efficiency of extraction. Recently, researcher report about 10-fold higher yields of fucose-sulphated polysaccharides (FBPs) extracts from A. nodosum using MAE, although antioxidant activities of extracts were slightly improved using ultrasound extraction technique. Furthermore, the simultaneous application of microwave and ultrasounds causes significant yields increasement of extraction of FBPs compared to its single utilization and further confirmation is in need to test the antioxidant properties of these extracts in other biological models [13].
For section 2.3. Seaweed contains not only astaxanthin but other fat-soluble vitamins as well. Complete the section with relevant information. Supplement Table 1 with the dose / concentration studied, positive / negative controls. The effect of astaxanthin on cerebrovascular disease is important, but it has other effects in a variety of diseases. Perhaps this section should be modified into a separate manuscript, and this manuscript should consider the various fat-soluble vitamins.
Response: thank you for the suggestion. We still prefer to keep section 2.3 in this manuscript because the section is not enough to be a new manuscript.
For section 2.4. Supplement this section with more detailed information on preclinical and especially clinical beta-glucan data.
Response: Thanks a lot. The following discussions were provided:
Nakashima and colleges report that paramylon isolated from E. gracilis Z exerts an immunoregulatory role in protecting against influenza virus infection in mice [15]. Oral administration of paramylon results in the production of higher amount of IL-1β, IL-6, IL-12 (p70), IFN-γ, IL-10, and IFN-β and significantly lower virus titer in the lung, indicating paramylon serves to alleviate symptoms of influenza virus infection [15]. E. gracilis paramylon treatment induces the production of NO, TNF-α and IL-6 through activation of the NF-κB and mitogen-activated protein kinase (MAPK) signaling pathways and thus activate the immune system in murine RAW264.7 cells [16]. Laminarin-type β-(1→3)-glucan from Sargassum henslowianum functions in regulating the intestinal microbiota composition by stimulating the growth of species belonging to Enterobacteriaceae while depleting Haemophilus parainfluenzae and Gemmiger formicilis [17]. Supplementation with β-1,3-glucan from E. gracilis may reduce and prevent upper respiratory tract infection (URTI) symptoms in human, including fewer sick days, URTI symptoms, URTI symptom days, URTI episodes, and lower global severity [18].
For section 2. Include an additional section on vaccines and sera from microalgae.
Response: Thanks a lot. Section 3 is about vaccines from microalgae.
For section 3. Supplement the corresponding section with the achievements and difficulties in the production of beta-glucan, fat-soluble vitamins, and sulfated polysaccharides.
A key aspect of the active compounds is their bioavailability. Consider the problem of absorption of microalgae-derived pharmaceuticals (e.g. https://doi.org/10.3390/md18110557).
Response: Thank you very much for the wonderful paper.
Production of beta-glucan, fat-soluble vitamins, and sulfated polysaccharide enable the application of algae for human health or medical field. Increasement on glucan content can be achieved through optimize growth conditions. For instance, nitrate starvation enhanced the β-glucan content of S. ovalternus SAG 52.80 from 23.3% to 36.7% and of Scenedesmus obtusiusculus A 189 from 16% to 23%, while increased irradiance enhanced the β-glucan content of S. obtusiusculus A 189 from 6.4% to 19.5% [19]. Algal biomass and bioproduct content are critical and careful considerations need to be considered to keep the balance. Although heterotrophic (HT) cultivation derived 4-times more of algal cell dry weight than photoautotrophic (PT) cultivation [20], PT cultivation leads to a production of more antioxidants than HT and mixotrophic cultivation, but similar free amino acid in Euglena gracilis [21]. In addition, industrial-scale heterotrophic cultivation of Eu. gracilis is limited by the high cost of medium containing organic carbon source like glucose, vitamins (especially B1 and B12), medium sterilization, and cultivation environments to avoid contaminants [22, 23]. Complex medium based on potato liquor from waste stream of the potato starch industry [23] might be a cheaper alternate. Development on biotechnological techniques ie. genetical modification or metabolic engineering of algal strain show great promise for enhanced algal biomass/bioproduct yields, although considerations need to be taken as discussed in [22, 24] .
We have also noticed the pharmacokinetic issue of bioactive compounds derived from algae. The bioactive compounds like Beta-glucan (paramylon), fucoidan and Griffithsin have a great molecular weight (>500), which means these compounds do not obey both Lipinski’s and Veber’s rules of drug-likeness and do not possess high degrees of predicted bioavailability. In other word, they do not have drug-likeness, or it’s very hard to make them into drug or they have a low absorption. But astaxanthin can pass through BBB, a few of algae-derived molecules can do this [25].
Reviewer 2 Report
Peer review report on “Recent Advances and Future Perspectives of Microalgae-Derived Pharmaceuticals”.
Manuscript ID: marinedrugs-1458362
Microalgae bioactive compound production possesses the potential to address the need for a wide range of pharmaceuticals products that are difficult or too expensive to be produced from other sources. Their Generally Recognized as Safe (GRAS) status and ability to be farmed for fast large-scale production makes this an attractive target for ecologically sound, commercially viable business opportunities in the future. In addition to dietary nutraceuticals, research has been focused on the investigation of the production and benefits of PUFAs, polysaccharides, astaxanthin, and recombinant proteins such as vaccines from these organisms.
This paper is generally well-written and presented and researched adequately with up-to-date references. The content is topical and will be of interest and significance to many readers.
Some comments:
Line 77: g-linolenic acid should be α-linolenic acid.
Line 88: Write as “Sufficient intake of arachidonic acid (ARA)…..”
Line 90: This is confusing. When is a substance both a vasoconstrictor and a vasodilator? Please clarify.
Line 91: Please rewrite. Neutrophils do not have an endothelium as this implies.
Figure 2: “Pyruate” should be “Pyruvate”.
Line 245: “neutrons” should be “neurons”.
Line 287: Reference 35 is incorrect with regards to the previous sentence.
Line 356-359: This sentence makes no sense. Please rewrite.
Author Response
Comments from Reviewer 2
Microalgae bioactive compound production possesses the potential to address the need for a wide range of pharmaceuticals products that are difficult or too expensive to be produced from other sources. Their Generally Recognized as Safe (GRAS) status and ability to be farmed for fast large-scale production makes this an attractive target for ecologically sound, commercially viable business opportunities in the future. In addition to dietary nutraceuticals, research has been focused on the investigation of the production and benefits of PUFAs, polysaccharides, astaxanthin, and recombinant proteins such as vaccines from these organisms.
This paper is generally well-written and presented and researched adequately with up-to-date references. The content is topical and will be of interest and significance to many readers.
Some comments:
Line 77: g-linolenic acid should be α-linolenic acid.
Line 88: Write as “Sufficient intake of arachidonic acid (ARA)…..”
Line 245: “neutrons” should be “neurons”.
Response: Thanks. The mentioned words were changed.
Line 90: This is confusing. When is a substance both a vasoconstrictor and a vasodilator? Please clarify.
Response: Sorry for the negligence. ARA and EPA are vasodilators, which serve to function to depress the function of vasoconstrictors. We have deleted vasoconstrictors.
Line 91: Please rewrite. Neutrophils do not have an endothelium as this implies.
Figure 2: “Pyruate” should be “Pyruvate”.
Response: we have rewritten them, thank you.
Line 287: Reference 35 is incorrect with regards to the previous sentence.
Response: Sorry for the mistake. The reference for “astaxanthin treatment significantly decreased cecal ligation and puncture-induced lung damage, as well as the resulting mortality rate in rats. The putative mechanisms underlying the protective effects against lung injury suppress inflammatory responses and inhibit NF- κBP65 expression” is supplied [26].
Nagendraprabhu P, Sudhandiran G. Astaxanthin inhibits tumor invasion by decreasing extracellular matrix production and induces apoptosis in experimental rat colon carcinogenesis by modulating the expressions of ERK-2, NFkB and COX-2. Invest New Drugs. 2011 Apr;29(2):207-24.
Line 356-359: This sentence makes no sense. Please rewrite.
Response: The sentence has been rewritten. “Moreover, Shahriari et al reported the successful construction of anti-newcastle disease virus recombinant subunit vaccines in microalgae C. reinhardtii platform through an agrobacterium-mediated genetic transformation system.”
Round 2
Reviewer 1 Report
The authors made the necessary corrections and I have no more questions.